# Acetylation, Phosphorylation, Ubiquitination (Oh My!): Following Post-Translational Modifications on the Ubiquitin Road

**DOI:** 10.3390/biom12030467

**Published:** 2022-03-18

**Authors:** Rachel E. Lacoursiere, Dania Hadi, Gary S. Shaw

**Affiliations:** Department of Biochemistry, University of Western Ontario, London, ON N6A 5C1, Canada; rlacours@uwo.ca (R.E.L.); dhadi@uwo.ca (D.H.)

**Keywords:** ubiquitination, acetylation, phosphorylation, protein structure, proteomics, cancer, neurodegenerative disease

## Abstract

Ubiquitination is controlled by a series of E1, E2, and E3 enzymes that can ligate ubiquitin to cellular proteins and dictate the turnover of a substrate and the outcome of signalling events such as DNA damage repair and cell cycle. This process is complex due to the combinatorial power of ~35 E2 and ~1000 E3 enzymes involved and the multiple lysine residues on ubiquitin that can be used to assemble polyubiquitin chains. Recently, mass spectrometric methods have identified that most enzymes in the ubiquitination cascade can be further modified through acetylation or phosphorylation under particular cellular conditions and altered modifications have been noted in different cancers and neurodegenerative diseases. This review provides a cohesive summary of ubiquitination, acetylation, and phosphorylation sites in ubiquitin, the human E1 enzyme UBA1, all E2 enzymes, and some representative E3 enzymes. The potential impacts these post-translational modifications might have on each protein function are highlighted, as well as the observations from human disease.

## 1. Introduction

As a prominent post-translational modification (PTM), ubiquitination controls a multitude of signalling events inside cells including proteasomal degradation, DNA damage repair, cell cycle progression, and more. A series of three enzymes (the E1 activating enzyme, an E2 conjugating enzyme, and an E3 ligase) sequentially bind and transfer the small modifying protein ubiquitin (Ub) onto lysine residues within substrate proteins. In humans, two Ub-specific E1 proteins exist, ~35 Ub-specific E2 conjugating enzymes, and nearly 1000 identified E3 ligases [1]. These enzymes are tightly regulated and sometimes require external switches to control their function.

To further modulate protein function, other amino acid side chains can be post-translationally modified to influence protein folding, domain organization, or protein–protein interactions. In addition to ubiquitination, protein modifications such as acetylation, methylation, or phosphorylation occur at thousands of individual protein sites and are key to the regulation of processes such as transcription, metabolism, trafficking, or proteostasis [2,3,4,5,6,7]. The development of high-throughput mass spectrometry experiments has now enabled the detection of these PTM sites [8,9,10]. Typically, these experiments provide full proteome data for a given PTM under a specific type of cell stress or treatment. This approach has now yielded thousands of PTMs that warrant further investigation under more targeted conditions.

Further to the general observations made from high-throughput experiments, certain sites can be modified by more than one type of PTM. For example, the acetylation and methylation of basic residues (lysine and/or arginine) acts to neutralize the side chain, creating a residue that is larger and reduces overall polarity. Ubiquitination also primarily modifies substrate lysine residues, indicating there may be complex crosstalk between multiple PTMs in cells. In addition to the competition for a single target residue by multiple types of PTMs, complexity is increased by the fact that multiple different PTM pathways can be used to initiate similar outcomes. This is most obvious in kinase signalling cascades, such as the activation of mTOR complex 1 (mTORC1) that can occur through synergistic PI3K/AKT or MEK/ERK signalling [11]. Furthermore, a protein can be ubiquitinated by multiple E2:E3 combinations yielding tens of thousands of E2:E3 pairs available to target for the same cellular fate. For example, p53, a central protein in cell cycle control, is ubiquitinated by the E3 ligases MDM2 [12,13], MKRN1 [14], TRIM24 [15], MUL1 [16], and HRD1 [17]. Ubiquitination of p53 by any of these E3 ligases with the E2 conjugating enzyme UBE2D1 results in the destabilization of p53 protein levels through degradation.

Similarly, there are around 40 lysine acetyltransferases (KATs), found in both nuclear and cytoplasmic compartments, that function to acetylate lysine sidechains [18]. Additionally, approximately 20 deacetylases that reverse acetylation are often targets for therapeutic intervention [19,20], though these are mostly poorly characterized. The small number of enzymes for acetylation/deacetylation compared to those for phosphorylation and ubiquitination implies the necessity for overlapping function and poor substrate specificity to account for the thousands of protein lysine acetylation sites observed. Future identification of new enzymes with uncharacterized KAT or deacetylase activity would provide more detail into the regulation of various proteins through acetylation.

The tight regulation of protein levels and their functions by PTMs is crucial to maintaining normal cellular activity. Aberrant PTMs are now regarded as a key hallmark in many diseases. For example, phosphorylation has been implicated in the development of breast cancer [21,22,23], Parkinson’s [24,25], and Alzheimer’s [26,27,28] diseases. The dysregulation of acetylation pathways has ties to prostate and lung cancers [29,30], and Alzheimer’s disease [31,32]. For ubiquitination, the modulated expression of numerous enzymes has been observed in multiple forms of cancer including prostate, colorectal, and pancreatic cancers [33,34]. Additionally, ubiquitination is involved in immune [35], developmental [36], and neurological disorders [37,38].

With the hundreds of proteins involved in the ubiquitination pathway, how is it that these E2 and E3 proteins know where and when to localize to maintain proper function? How can E2:E3 pairs actively discriminate for one or more substrates at a given time, and what exactly causes a given E2:E3 pair to modify their innate polyubiquitin chain building preference to change the fate of the modified substrate? One way to modify these events is through post-translational modification of the E2 and E3 enzymes. This would add an additional level of regulation and place a tighter control on the activity of enzymes in the ubiquitination pathway. One of the best examples of this is the E3 ligase parkin where the alleviation of parkin autoinhibition by S65 phosphorylation and the non-covalent binding of phospho-ubiquitin is required for maximum ubiquitination activity [39,40,41]. In addition to parkin, other E3 ligases including BRCA1 [42,43,44] and ITCH [45,46,47], undergo post-translational modification to regulate their functions. Many proteomics studies have now identified E2 conjugating enzymes and E3 ligases that are phosphorylated or acetylated under a variety of conditions. Furthermore, lower-throughput experiments in the literature detail the involvement of these modified ubiquitination proteins in downstream events. Here, we review PTMs to proteins in the ubiquitination pathway. We have compiled the PTMs that occur in ubiquitin, UBA1 (E1) and E2 conjugating enzymes, along with selected E3 ligase enzymes, to show how PTMs might alter their structures and interactions and impact downstream ubiquitination. We focus on what cellular events cause the modification of these proteins and their general observations in disease.

## 2. Ubiquitin

The small modifier ubiquitin (Ub) is an 8.5 kDa protein that contains seven lysine residues (K6, K11, K27, K29, K33, K48, K63) and an N-terminus that can be modified through ubiquitination. When ligated to a substrate protein, the type of linkage and length of the chain provide signals for proteasomal or lysosomal degradation, trafficking, and signalling. Six of these lysine residues (not K29) have been observed to be acetylated (Table 1, Figure 1). For ubiquitination, the abundance of cellular pathways and outcomes influenced by the formation of polyubiquitin chains on Ub itself indicates that disease correlations are complex. Alterations in the levels of ubiquitination at each of the lysine residues in Ub have been observed in colorectal cancer and leukemia, indicating that global ubiquitination is upregulated in these diseases, rather than a specific form. The outcomes of various topologies of polyubiquitin chains have been extensively reviewed, and readers are invited to read references [35,48,49] for details and specific roles in disease. Not surprisingly, the acetylation of different lysine residues in ubiquitin prior to passage through the E1, E2, and E3 enzyme cascade has been shown to eliminate the formation of specific polyubiquitin chain types. For example, the acetylation of Ub at K6 or K48 impairs the formation of polyubiquitin chains at K11, K48, or K63 in vitro [50]. Furthermore, in vitro ubiquitination experiments demonstrate that Ub acetylation modulates the rate of formation for E2~Ub conjugates [51]. Yet, the point when acetylation of Ub occurs is unclear. Current proteomics methods have been unable to discriminate between unanchored and in-chain acetylated Ub, which adds challenges in addressing this question. The detection of basal levels of acetylated K6 and K48 Ub indicates that some of these modifications could arise prior to Ub passage through the ubiquitination cascade [50]. The acetylation of Ub is upregulated due to DNA damage, the induction of autophagy, or deacetylase inhibition, but only K6 acetylation is detected in patient-derived leukemia cell line K562 and non-small cell lung cancer A549 cells [52].

All serine (S20, S57, S65) residues, six of seven threonine (T7, T12, T14, T22, T55, T66) residues and the single tyrosine (Y59) have been identified as phosphorylation sites in Ub. As with acetylation, increased levels of these modifications are observed following particular cellular events including oxidative stress (S65) [41] or DNA damage (T12) [92]. It is unknown which kinases regulate the phosphorylation of most sites in Ub. However, the phosphorylation of S65 by PINK1 is particularly well described, triggered by oxidative stress following substrate ubiquitination at the outer mitochondrial membrane [39,93]. This event is essential for translocation and the activation of the E3 ligase parkin. In vitro phosphorylation of other serine/threonine residues in Ub fail to stimulate parkin E3 ligase activity at similar levels [94]. Recently, phosphorylation at S57 of Ub by the kinases Sks1/Vhs1 (yeast) or MARC2 (human) has been described [95]. Phosphorylation of Ub appears to be more abundant than acetylation in disease tissue. Phosphorylation at T7, T12, S20, T22, S57, Y59, and S65 have all been detected in various tumors and cancer-derived cell lines [53,54]. The lack of clearly identified PTM pathways responsible for modifying Ub limits the interpretation of many of these results. It would be expected that other forms of modified Ub serve regulatory roles in various ubiquitination cascades similar to pS65 Ub in the Parkin/PINK1 mitophagy pathway.

## 3. E1 Activating Enzyme (UBA1)

The most prevalent human Ub-specific E1, UBA1, is comprised of a single polypeptide organized into five structural domains and a bundle of helices (4HB) (Figure 2a). Two of these domains, the active adenylation (AAD) and the second catalytic cysteine half (SCCH) domains, contain catalytic functions to adenylate and conjugate a Ub molecule, respectively. In the AAD, a molecule of ATP is coordinated by A478, D504, R515, and K528, which enable interactions with ATP-binding residues D576 and N577 (residue numbering refers to human UBA1, PDB 6DC6). In the SCCH domain, the catalytic cysteine (C632) is located on the anterior surface of the domain, over 30 Å from the adenylation pocket. Two other domains, the inactive adenylation (IAD) and the first catalytic cysteine half (FCCH) domains, stabilize the structure of UBA1 and have roles in positioning the SCCH domain. An accessory domain, the ubiquitin fold domain (UFD), is located at the C-terminus of UBA1 and, together with the SCCH domain, has a crucial role in E2 recruitment [96,97].

During catalysis, the UBA1 protein undergoes dynamic rearrangement to sequentially adenylate the C-terminus of Ub, form the covalent thioester-linked UBA1~Ub intermediate, and transfer the Ub onto the catalytic cysteine of an E2 protein through a transthiolation reaction. These functions require the SCCH domain to adopt two conformations: (1) an open conformation [98,99,102,103,104] where adenylation and transthiolation are favoured and UBA1 C632 is distal to the adenylation site; and (2) a closed conformation [105,106] where the SCCH domain folds and rotates backwards to draw C632 into proximity of the adenylation site to favour thioesterfication. The successful rearrangement from open to closed and back requires the SCCH domain to adjust contacts with both the AAD and FCCH domains. Functional UBA1 mutagenesis experiments have highlighted these interactions and show that suboptimal domain–domain interactions result in decreased enzymatic function [98,99,106]. Similarly, it is clear that post-translational modifications of UBA1 that hinder the recruitment of ATP or an E2 enzyme, or alter domain–domain interactions, might be expected to impact enzymatic activity.

Mass spectrometry experiments have revealed that a very large number of residues in all domains of UBA1 can be post-translationally modified by acetylation, phosphorylation, or ubiquitination (Table 1, Figure 2b,c). The highest density of PTMs appears to be in the SCCH domain, where 12 lysine residues can be modified by either acetylation or ubiquitination (K627, K635, K657, K671, K746, K802, K806, K830, K838, K843, K884, K889) and 15 residues show evidence of phosphorylation (S628, T682, S781, T789, S793, T800, S803, S810, S816, S820, S824, S835, S855, S866, Y873). Although many PTM sites are on the exterior surface of the protein, multiple modifications lie in the catalytically active AAD (Q448-D623, I891-A942) and SCCH (P624-I890) domains, providing clues that these PTMs impart functional changes observed in diseases such as breast cancer, leukemia, and multiple myeloma (for information on disease states and cell lines, readers should refer to references contained in Table 1 and PhosphositePlus^®^ [107]). Ubiquitination of K528 in the AAD, as observed in multiple myeloma cell lines, has obvious effects on the downstream cascade. K528 is one of four residues that coordinates the ATP molecule required for Ub activation [102]; neutralization of the K528 sidechain through acetylation or the addition of a bulky Ub protein would be expected to eliminate ATP binding, rendering the catalytic activity of K528-modified UBA1 minimal. Previous kinetic work has also shown that K528 is crucial to the ordered binding of substrates, where ATP binding precedes Ub recruitment [108]. Disruption of ATP binding might also be affected by acetylation of K884 (SCCH). This residue is hydrogen bonded to D852 near the SCCH, IAD, AAD interface [102]. Its analogous interaction in *Schizosaccharomyces pombe* (*S. pombe*, *sp*) UBA1 shows *sp*K845 (K884) positions *sp*D813 (D852) near the ATP-coordinating residue *sp*R22 in the UBA1 open conformation. Charge substitution of *sp*D813 reduces the formation of E1~Ub by 80% [105] due to the inability of *sp*R22 to position an ATP molecule [98]. It might be expected that the acetylation of K884 found in gastric or lung cancers might indirectly modulate ubiquitination due to the trickle-down effects resulting from the altered positioning of R57 (*sp*R22) in the K884-D852-R57-ATP hydrogen bonding network. Similarly, the ubiquitination or acetylation of K604 or K923 (AAD) would likely alter Ub adenylation, since these residues lie in the same region as F320, F926, and F933 that comprise a hydrophobic surface where the adenylated Ub molecule sits during activation [102]. Ubiquitination of K923 is detected in leukemia and multiple myeloma cells and would be expected to disrupt the positioning of the I44 patch of Ub.

There are multiple PTM sites in the SCCH domain that lie near the catalytic cysteine (C632). These include the ubiquitination sites K635 and K746 observed in leukemia and multiple myeloma and the Y873 phosphorylation site observed in breast and liver cancers. In particular, K635 and K746 sit on either side of C632, while Y873 projects towards the catalytic loop. Together, these residues that are subject to PTMs sit near key residues that control the open/closed forms of UBA1 and E2 recruitment. Structures of *S. pombe* UBA1 in complex with the E2 protein Ubc15 reveal that *sp*K596 (homologous to K635) makes polar contacts with the incoming E2 protein (H84) to correctly position the catalytic cysteines of the E1 and E2 proteins and facilitate Ub transfer [100]. Nearby, the sidechain of *sp*K706 (K746) coordinates the D133/S135 loop in Ubc15 [100]. The sidechain of *sp*Y834 (Y873) extends towards the catalytic cysteine, forming a hydrogen bond with the backbone carbonyl of *sp*N597 (N636) near *sp*C593 (C632) that is conserved in *S. pombe* UBA1 structures [106]. The phosphorylation of Y873 in human UBA1 observed in breast and liver cancers would introduce a highly negative residue near the FCCH domain, modifying the dynamics between the open and closed state to promote a closed state. The promotion of a closed state would decrease the amount of Ub activation and the amount of E2~Ub formed over time, resulting in poorer catalytic activity of UBA1. The sites of these three PTMs are very close to the *sp*F598, *sp*F689, and *sp*F701 (F637, F729, F741) hydrophobic patch in the SCCH domain required for E2 recruitment by UBA1 [98,100,106]. This hydrophobic region of UBA1 has been well studied with other E2 proteins including Ubc4 [98] and Cdc34 [99]. Furthermore, interactions between *sp*R707 (K746) and *sp*E214 (FCCH domain) maintain the UBA1 in an open conformation, although these interactions are less obvious in the human enzyme. Replacing the basic *sp*R707 with acidic *sp*E707 results in charge repulsion between the FCCH and SCCH domains and correlates to a 75% reduction in E2~Ub formation due to promoting the closed conformation [106]. The ubiquitination of either K635 or K746 would disrupt E2 recruitment by multiple mechanisms including blocking the phenylalanine triad or elimination of the hydrogen bonding network between K746 and the D133/S135 loop in the E2 enzyme. This latter suggestion is supported by mutagenesis experiments that show that S135A substitution in Ubc15 severely compromises the rate of transthiolation from UBA1 [100]. This interaction is also expected for the acetylation of K746 that is also observed. 

The UBA1 protein is large and requires a coordinated effort to activate and transfer Ub to an E2 enzyme. The potential effects of post-translational modifications to UBA1 stem from a multitude of factors: (1) the domain in which the residue is located, (2) nearby structural contacts or surfaces favored in either the open or closed state, and (3) the nature of modification (bulky such as ubiquitination, charge introduction such as phosphorylation, or charge neutralization such as acetylation). While a handful of these sites provide compelling structural evidence as to how the modification might alter downstream ubiquitination, many others do not have obvious structural effects. It remains unclear which PTM sites in UBA1 act as positive and negative regulators in downstream ubiquitination. Further work into uncovering the effects of these PTM sites will provide evidence as to which sites result in the loss or gain of function and will aid in understanding the molecular mechanism of diseases such as leukemia and other cancers.

## 4. E2 Conjugating Enzymes

All E2 enzymes contain a 150-residue catalytic domain (UBC, Figure 3), sometimes accompanied by additional N- or C-terminal regions that can influence ubiquitin handling [109]. The UBC domain contains an α/β fold with its catalytic cysteine positioned in a pocket below the α2 helix (crossover helix). On either side of the catalytic pocket lie residues positioned to facilitate nucleophilic attack by either a lysine residue in conjunction with RING E3 ligases or a cysteine residue for HECT and RBR E3 ligases. E2 enzymes contain an HPN motif that is necessary to accept Ub from UBA1, positioned near the catalytic cysteine. The transfer of Ub requires recruitment of one of about 35 possible E2 enzymes by the E1 enzyme, UBA1, transthiolation of Ub to the E2 enzyme, and the subsequent release of the E2~Ub conjugate. Generally speaking, three-dimensional structures show that the E1:E2 interaction utilizes the N-terminus of helix α1 and the L4 and L7 loops of the E2 enzyme (Figure 3) that interact with the UFD and AAD regions of the E1 enzyme (Figure 2). Supporting interactions occur between the base of the E2 protein that includes the HPN motif and catalytic cysteine and the N-terminus of helix α3 that both face the SCCH domain of UBA1 [98,99,100]. Similarly, crystal structures of the E2 enzymes in complex with either RING or HECT E3 ligases reveal that similar regions of helix α1 and L4 and L7 loops are important [110,111,112,113,114,115]. The architecture and importance of particular E2 enzymes and their biological roles have been extensively reviewed [116,117].

Post-translational modifications have been identified in the UBC domains of 33 different E2 conjugating enzymes (Table 1). Several E2 enzymes, such as UBE2R1/UBE2R2 and UBE2K, have multiple phosphorylation, acetylation, and ubiquitination sites located outside their UBC domains that are not highlighted in this review. It is unclear at what point these E2 enzymes are modified, and which acetyl transferases and kinases are involved. However, it is apparent that PTMs that occur prior to involvement in the ubiquitination cascade could significantly impact ubiquitin transfer. Analysis of the PTMs reveals that there are very few “hotspots” in the E2 proteins, likely due to the poor conservation of lysine, serine, and threonine residues through the E2 sequences. Furthermore, PTMs are found infrequently in the L4 and L7 loop regions needed for E1 or E3 recruitment, although phosphorylation has been noted in L4 of UBE2R1 (S71) and UBE2S (S73) and in L7 of UBE2D2 (S94, T98). Among many disease-relevant systems, these L4 or L7 phosphorylation sites have been observed in cervical adenocarcinoma HeLa cells, PC9 lung cancer cells, and primary tissue from thyroid cancer and gastric carcinoma [10,53,55,56,57,119]. One notable exception to the lack of “hotspots” is K8 (UBE2D family numbering) in helix α1 that can be acetylated or ubiquitinated in UBE2D2, UBE2D3, UBE2D4, UBE2E1, UBE2L3, UBE2L6, UBE2N, and UBE2S. PTMs at this position occur in a range of disease states, including colorectal cancer tissue, multiple myeloma L363, KMS27, and RPMI-8266 cells, leukemia MV4-11 and K562 cells, lymphoma SU-DHL-4 cells, and many other cancer-derived cell lines [8,58,59]. In some E1:E2 structures this residue sits near acidic residues at the UFD interface that would be disrupted through acetylation or steric hindrance of ubiquitination. Substitution of the analogous residue in Ubc15 or Ubc12 leads to significant decreases in E2~Ub thioester formation [100,101]. A K8 hydrogen bond has also been noted in the complex with the HECT E3 ligase NEDD4L [114] and for K9 of UBE2L3 with RBR E3 ligases such as parkin and ARIH1 [120,121]. Modification at this lysine position may prove influential, since UBE2L3 and the UBE2D family of E2 enzymes are specifically utilized by HECT and RBR E3 ligases where Ub is transferred to a catalytic cysteine on the acceptor E3 enzyme. Structures of UBE2L3 with either parkin or ARIH1 consistently show proximity to T242/D243 (parkin) or hydrogen bonding to the backbone CO from Y190 (ARIH1) [120,121]. Interestingly, a D243N substitution in parkin is causative for early onset Parkinson’s disease [122]. This structural observation likely does not apply to most other E2 enzymes which carry alternative residues (R, A, L) at this position that would preclude modification. The highest density of PTMs for the E2 enzymes appears in helix α4, where K144 in the UBE2D family is consistently acetylated or ubiquitinated in various cell lines related to prostate cancer, gastric carcinoma, and leukemia [9]. The downstream effects of K144 modification are unclear, since this region has not been highlighted in interactions with either E1 or E3 enzymes. Interestingly, the UBE2D family of E2 proteins non-covalently coordinates an additional Ub to facilitate the formation of polyubiquitin chains [123,124,125]. The positioning of this Ub is observed in multiple structures [113,124,126] where the I44 hydrophobic patch of Ub packs against β1-3 of the E2 enzyme. This binding site positions the C-terminus of Ub in proximity to the C-terminus of helix α4 of the E2 protein, which contains K144. It might be expected that the ubiquitination of the UBE2D family of E2 proteins at K144, as observed in various cancer-derived cell lines, would occupy a similar position to that observed in non-covalently bound Ub:UBE2D structures. UBE2D ubiquitinated at K144 would likely prevent the oligomerization of UBE2D~Ub conjugates, stalling downstream ubiquitination machinery. Another E2 protein capable of non-covalently binding a Ub molecule along this backside surface is UBE2G2 [127]. Ubiquitination at the UBE2G2 site analogous to K144 (K161) has also been observed in K562 leukemia cells.

PTMs might influence the position of the covalently attached Ub protein in an E2~Ub conjugate. It is well established that the thioester-linked Ub can occupy either an open or closed conformation. In the closed state, demonstrated for Ubc1, Cdc34, and UBE2L3 in solution, the I44 patch of Ub organizes against the crossover helix of the E2 enzyme [128,129,130] and is a requirement for Ub unloading. This conformation is observed in structures of E2~Ub conjugates with RING E3 ligases such as RNF4 with UBE2D1~Ub [131]. In this closed conformation, K101 near the N-terminus of the UBE2D1 crossover helix is wedged between L8 and the β5 strand of Ub making a hydrogen bond to the backbone carbonyl of T7 in Ub. Substitution to K101A in UBE2D1 leads to reduced Ub transfer in single turnover assays with RNF4 [131]. While the ubiquitination of K101 has not been observed in UBE2D1, it has been found in UBE2D2 and UBE2D3, and would be expected to dampen this interaction. In the open conformation of the E2~Ub conjugate, the Ub has limited interactions with the UBC domain, instead occupying a range of positions [132,133,134,135]. It would be expected that PTMs in the E2 enzyme would have a smaller impact on the Ub position, and subsequent Ub transfer, in these cases.

Overall, there are a large number of PTMs observed in E2 enzymes. Most of these appear to be located in regions outside those normally involved with other enzymes in the ubiquitination cascade. Furthermore, some sites of ubiquitination might arise from the turnover of specific E2 enzymes needed to modulate their concentrations. Nevertheless, many of the PTMs are observed in various cancers and other diseases.

## 5. E3 Ligases

E3 ligases comprise nearly 1000 members that fall roughly into three categories. RING E3 proteins are abundant in cells and make up nearly 70% of the total number of E3 ligases. These ~60-residue domains comprise two Zn^2+^-binding sites and typically reside in proteins with multiple other domains. RING E3 ligases recruit an E2~Ub conjugate using two loops (L1, L2) that are parts of the Zn^2+^-binding motif and transfer ubiquitin directly to a substrate [136,137]. The family of HECT E3 ligases contains approximately 30 members [138]. These proteins usually contain several other regions or domains to facilitate interactions with other cellular components [139]. The catalytic HECT domain lies C-terminal to other domains and folds into a distinct structure comprised of the N-lobe (E2~Ub binding) and the C-lobe (catalytic cysteine containing). HECT E3 ligases transfer ubiquitin from the E2~Ub conjugate to the catalytic cysteine in the C-lobe. Finally, RBR E3 ligases are a smaller class of E3 proteins that utilize a hybrid mechanism of ubiquitin transfer that includes aspects of both the RING and HECT enzymes [140]. RBR proteins such as parkin, ARIH1, and HOIP contain a canonical RING domain required for E2~Ub recruitment, and transfer ubiquitin to a catalytic cysteine in their Rcat (RING2) domains prior to labelling a substrate [141]. The large number of E3 ligases and their complexity makes general statements about the locations or dominance of particular PTMs difficult. In lieu of this, we describe the impact of PTMs with selected E3 ligases from each class.

### 5.1. RING E3 Ligases

RING E3 ligases have roles in DNA damage repair, cell cycle regulation, and signalling events. Among the ~600 RING E3 ligases are a number that act as proto-oncogenes, controlled by genetic mutations or epigenetic regulation. Additionally, the functions of proto-oncogenic RING E3 ligases such as MDM2 and CBL are regulated through post-translational modification of the ligase.

Human MDM2 is a well characterized proto-oncogenic RING E3 ligase that modulates the transcription factor activity of tumor suppressor p53 during the cell cycle. Along with other E3 ligases, MDM2 adjusts p53 protein levels by ubiquitination and proteasomal degradation. This process is finely tuned through PTMs to MDM2. While no phosphorylation sites have been identified in the MDM2 RING domain, multiple phosphorylation sites have been observed just prior to the RING sequence that inhibit oligomerization of MDM2 and suppress p53 ubiquitination [142]. Additionally, the phosphorylation of S17 near the N-terminus of MDM2 alters p53 binding [143]. This PTM, detected in colorectal and lung cancers, increases discharge of the E2~Ub thioester in the absence or presence of a substrate, and increases proteasomal turnover of MDM2 [144]. Furthermore, an S17D substitution increases p53 binding and appears to activate the ligase function of MDM2 [145], providing evidence that pS17 upregulates the ubiquitination of p53 by MDM2. Another phosphosite in MDM2, S166, is required for the nuclear localization of MDM2 [146,147], necessary for the ubiquitination and degradation of p53 [146]. The reduced p53 levels corresponds to reduced sensitivity to etoposide-induced apoptosis [148], providing evidence that etoposide is less effective as a chemotherapeutic against cancers that have high levels of pS166 MDM2. In the absence of phosphorylation, MDM2 is readily degraded [149], stabilizing p53 levels and conferring sensitivity to etoposide [147]. Two key acetylation sites (K182, K185) in the nuclear localization sequence of MDM2 have also been identified. Acetylation of these residues by CREB-binding protein (CBP) enhances recruitment of the deubiquitinase HAUSP, counteracting MDM2 degradation through autoubiquitination, and facilitates p53 ubiquitination by the E3 ligase [150]. In addition, the acetylation of K466 in the RING domain has also been observed. Although this PTM impairs p53 ubiquitination, it is not clear how this occurs, since K466 is remote to the E2~Ub binding site in MDM2 [151].

The RING E3 ligase CBL is another proto-oncogene involved in proteasomal degradation and the ubiquitination of proteins involved in receptor tyrosine kinase (RTK) signalling. In addition to ubiquitinating various RTKs, CBL is a substrate for RTK kinase activity. Many residues in CBL are phosphorylated, including Y371 (which lies just N-terminal to the RING domain), and residues towards the C-terminus in adaptor domains (Y700, Y731, and Y774). Oncogenic mutations of CBL at Y371 including point mutations or large deletions, are observed in various forms of leukemia and result in the loss of E3 ligase activity and the gain of adaptor signalling activity [152]. In non-oncogenic forms of CBL, Y371 partakes in autoinhibitory contacts with the RING domain. The phosphorylation of Y371 relieves autoinhibition, increasing the affinity for an E2~Ub conjugate [153]. The conformational changes associated with CBL pY371 are required for RTK ubiquitination, providing insight as to how oncogenic mutations at this residue might eliminate ligase activity. Various Y371 oncogenic mutations, including Y371H, have been detected in patient samples [154,155] and studied further [156,157]. These substitutions result in an upregulation in the phosphorylation of the C-terminal tyrosine residues Y700, Y731, and Y774. The phosphorylation of these residues regulates multiple signalling cascades including the PI3K pathway [158] and the recruitment of nucleotide exchange factors to cell membranes [156,159,160].

BRCA1, a RING-containing tumor suppressor E3 ligase and implicated in breast cancers, forms a heterodimeric RING complex with BARD1 in the nucleus of cells. Together, the BRCA1/BARD1 ligase has roles in DNA damage repair and gene transcription through histone ubiquitination. Both BRCA1 and BARD1 are subject to post-translational modification, where the phosphorylation of various residues in each protein have been detected in breast cancer tissue and leukemia cells. Acetylation at K50 in the RING domain of BRCA1 has been observed. This residue is proposed to have interactions with the E2~Ub conjugate [161], where acetylation may negate salt bridge formation with the E2 enzyme [162]. The phosphorylation of BRCA1 S114, T509, and S694 are among the residues most frequently detected in both high- and low-throughput experiments. Unfortunately, there is little structural information to provide insight as to why these BRCA1 residues are phosphorylated and the result on ligase activity: the available structures are limited to the RING domain or the C-terminal BRCT domains. However, functional low-throughput experiments have provided insight as to the downstream effects of these phosphorylation sites. The phosphorylation of S114 increases after cell treatment with hydroxyurea [44], an oral chemotherapeutic commonly used to treat types of leukemia [163]. The BRCA1 S114D phosphomimetic supported DNA replication fork protection, while alanine substitution did not [44], suggesting a protective role of BRCA1 pS114 in genome stability. T509 is located in the nuclear localization signal of BRCA1 and is phosphorylated by AKT downstream of PI3K. Breast cancer cells treated with heregulin (a growth factor stimulating certain receptor tyrosine kinases) showed an increase in pT509 [42]. Furthermore, the same cell stimulus resulted in an increase in the nuclear localization of BRCA1 [164], suggesting that pT509 is required for promoting the colocalization of BRCA1 and chromatin. Furthermore, the phosphorylation of S694 is also achieved through the kinase activity of AKT. In breast and ovarian cancer cells, pS694 is necessary for the stabilization of BRCA1 protein levels after hormone stimulation with estradiol or insulin-like growth factor 1 [43]. Although not in the ligase active domain of BRCA1, the phosphorylation of S114, T509, or S694 each demonstrate crucial roles to regulating the E3 activity of BRCA1. It remains unclear how, structurally, many of these PTMs modulate protein–protein interactions, since they frequently reside in unstructured regions that may only form distinct elements when in complex with the appropriate binding partners.

### 5.2. HECT Ligases

Nedd4-2 is a predominantly cytoplasmic HECT E3 ligase containing an N-terminal C2 domain, multiple WW domains, and the C-terminal catalytic HECT domain. Nedd4-2 is heavily involved in the ubiquitination of various ion transporters including the epithelial sodium channel (ENaC), sodium-chloride cotransporter (NCC), and the family of organic anion transporters (OATs). The phosphorylation of two residues in Nedd4-2 (S342 and S448) have been observed in numerous studies in response to hormones such as dexamethasone, insulin, and aldosterone [165,166,167,168]. The phosphorylation of these residues has been detected in leukemia and gastric, breast, and lung cancers. Despite their remote position from the HECT domain in Nedd4-2, alanine mutagenesis experiments of S342 and S448 demonstrate that the phosphorylation of these residues inhibits the ubiquitination of different ion transporters, preventing their internalization and subsequent degradation [166,168,169]. This inhibition is a result of the decreased binding affinity of phospho-Nedd4-2 for the transporters [166,170]. In addition to transporter ubiquitination, Nedd4-2 plays a role in terminating TGF-β signalling, where the ligase mediates turnover of pSMAD3 [171]. The phosphorylation of Nedd4-2 at S448 inhibits pSMAD3/Nedd4-2 binding, sustaining TGF-β signalling. Finally, both pS342 and pS448 facilitate the binding of Nedd4-2 to 14-3-3 proteins [172]. Phosphorylation-induced 14-3-3:Nedd4-2 interactions reduce the autoubiquitination of Nedd4-2, a direct result of modified contacts to the HECT domain that impairs catalysis [172].

Within the HECT domain of Nedd4-2, S795 and T903 have both been shown to be phosphorylated. S795 lies in the N-lobe and is peripheral to the E2 binding site, so might be expected to have minimal impact on ubiquitination. However, T903 lies ~7 Å from the catalytic cysteine, C942, in the C-lobe of the domain and extends towards a primarily hydrophobic pocket comprised of L879, F893, L896 [114]. T903 is phosphorylated in lung cancer and functional experiments indicate that its phosphorylation is required for Nedd4-2 to promote the turnover of ENaC [173,174]. This may indicate that phosphorylation at T903 might promote a C-lobe conformation that favors catalysis.

The HECT E3 ITCH belongs to the same family of HECT proteins as Nedd4-2. ITCH contains a C2 domain and multiple WW domains to facilitate substrate interaction. The E3 ligase is capable of autoubiquitination and has been observed to be phosphorylated in breast and gastric cancer and leukemia. In triple-negative breast cancer cells, the phosphorylation of S257 promotes the nuclear translocation of ITCH and the subsequent ubiquitination of histone H1.2. The ubiquitination of H1.2 by ITCH prevents the accumulation of 53BP1 foci, directly antagonizing the roles of RNF168/RNF8 in the DNA damage response [47]. The nuclear localization of ITCH, as upregulated in breast tumors, is a result of pS257, where nuclear ITCH prolongs transcriptional activity, influences cell replication, and promotes tumor progression.

ITCH-mediated ubiquitination of JunB, a proto-oncogene transcription factor, promotes JunB turnover. In stimulated T cells ITCH is phosphorylated at Y420 by Fyn kinase, and pY420 reduces the interaction between ITCH and JunB, stabilizing JunB protein levels [45]. Additional ITCH substrates Notch and SMAD2 experience similar effects, demonstrating that the phosphorylation of ITCH at Y420 decreases the ubiquitination of known ITCH substrates through impaired substrate recruitment. Furthermore, ITCH phosphorylation at S687 in the N-lobe of the HECT domain inhibits the recruitment of UBE2L3, terminating the normal function of ITCH in TNF and WNT/β-catenin signalling cascades [175,176]. These examples of ITCH phosphorylation provide evidence that ITCH regulation occurs both prior to and after formation of the E3~Ub covalent intermediate, and that ITCH phosphorylation impacts a variety of cellular outcomes linked to disease.

A recent study on the HECT E3 ligase SMURF2 detected the phosphorylation of S384 in cells treated with etoposide, a chemotherapeutic used to treat various cancers [177]. Interestingly, the cells had no basal level of pS384, indicating that phosphorylation of this residue is dependent on the induction of double-stranded breaks (DSBs) in DNA. At DSBs, SMURF2 ubiquitinates RNF20 to promote its degradation; removal of the phosphorylation site by alanine mutagenesis (S384A) decreases RNF20 ubiquitination. Low levels of RNF20 in the nucleus prevent heterodimerization with RNF40 and impair the repair of DSBs. Therefore, the treatment of cancers with etoposide is accomplished, in part, through the destabilization of the genome in a pS384 SMURF2-dependent manner.

An examination of the HECT E3 ligases reveals that numerous other PTM sites have been detected, although many of these lack low-throughput experiments to evaluate the consequences of the PTM on ligase activity. Additionally, it is clear that phosphorylation is a dominant PTM compared to acetylation in the HECT proteins. For example, multiple phosphorylation sites have been identified in all HECT E3 ligases, many of which contain multiple sites in the HECT domain. HECT proteins WWP1, ITCH, and HERC5 only have one known phosphorylation site in the HECT domain, while phosphorylation sites in the HECT domains of HECW1, HECTD2, and G2E3 have yet to be identified. However, acetylation has not been reported (or is extremely limited) for the NEDD4 subfamily of HECT E3 ligases and minimally in the HERC group (HERC2, HERC6). Other HECT E3 ligases, such as HUWE1, HACE1, and UBE3A (E6AP) appear to have greater acetylation tendencies. Confirmation of these PTMs and subsequent functional studies will be required to identify how PTMs might modify protein–protein interactions, substrate turnover, or cellular localization.

### 5.3. RBR Ligases

One of the best characterized proteins in the RBR E3 ligase family is parkin, a key player in mitochondrial homeostasis and implicated in Parkinson’s disease [178]. Although the phosphorylation of parkin has been identified for several residues in its Ubl (S9, S10, S19, S65), RING0 (Y143, S193, S198, T204), RING1 (S296) and IBR (Y372, S378) domains, there are no reports of lysine acetylation. Most of these PTMs appear to be peripheral to the Ubl auto-inhibition, phosphoubiquitin or UBE2L3 binding sites, and would not be expected to dramatically affect ubiquitination function. It is notable that substitutions at Y143 or S193 do result in rare forms of Parkinson’s disease [179,180]. In contrast, phosphorylation at S65 is a key step to parkin’s ubiquitination activity. Parkin activity is regulated by the mitochondrial kinase PINK1 [181] that phosphorylates ubiquitin, required for parkin translocation to the outer mitochondrial membrane, and S65 of parkin’s Ubl domain [182,183]. These events result in the ubiquitination of a wide range of mitochondrial proteins as a signal for mitophagy [184,185]. Additional regulation of the PINK1-parkin mitophagy pathway can occur through non-protein factors. For example, the treatment of bladder cancer EJ cells with the carcinogen antimony reduces pS65-parkin protein levels resulting in the inhibition of mitophagy and the increased invasiveness of cells [186]. Interestingly, sequential TNFα or IFNγ stimulation to induce the expression of the Ub-like protein FAT10 followed by mitochondrial decoupling causes a pS65-dependent increase in FAT10ylation, ultimately reducing parkin ubiquitination activity [187]. These opposing influences that S65 phosphorylation has on parkin ubiquitination indicates that there is a complex network of events that regulate parkin function, and that PINK1-mediated phosphorylation could increase or decrease parkin activity depending on other signals. Other phosphorylation sites such as S9 in the Ubl domain and Y143 in the RING0 domain have also been observed to regulate non-mitochondrial parkin function. For example, parkin ubiquitination of RIPK3 prevents its activation and downstream necroptosis, a process of inflammatory cell death that might contribute to colorectal tumors. RIPK3 ubiquitination is enhanced with the phospho-mimetic S9D parkin, providing evidence that parkin phosphorylation at S9 has a protective, tumor suppressor effect against necroptosis and tumor formation [188]. The phosphorylation of Y143 may exhibit opposite effects on parkin ligase activity, since c-Abl-mediated phosphorylation of parkin Y143 abolishes ubiquitination activity and results in the accumulation of parkin substrates AIMP2 and FBP1 [189]. Y143 phosphorylation is increased in the brains of Parkinson’s disease patients and has been found in mouse models stimulated with MPTP, a prodrug that causes Parkinson’s symptoms by killing dopaminergic neurons [190]. Heightened c-Abl activation has also been observed, demonstrating a link between Y143 phosphorylation and impaired parkin function [189].

For other RBR E3 ligases such as ARIH1 and HOIP, peptides corresponding to multiple acetylated, phosphorylated, and ubiquitinated residues have been detected in high throughput mass spectrometry experiments. HOIP ubiquitination at K1056 due to Toll-Like Receptor 4 stimulation impairs the linear chain building activity of the linear ubiquitin chain assembly complex (LUBAC) through conformational changes [191]. Unfortunately, these types of low-throughput experiments to study the effects of specific PTMs on ligase activity of RBR enzymes lack abundance. The development of chemical biology techniques and the identification of the modifying enzymes should enable the study of many more post-translationally modified proteins, including lesser well-studied RBR E3 ligases.

## 6. Conclusions

The functions of ubiquitination proteins in cells are tightly regulated through various events, including their post-translational modification with common events such as acetylation, phosphorylation, and ubiquitination. In many cases, large-scale proteomics studies detect various PTMs in E2 and E3 proteins in disease tissue, yet it remains unclear what upstream events cause their modification and the full extent of the downstream outcomes. Figure 4 shows three points in the ubiquitination cascade when proteins involved might undergo post-translational modification. First, any of the enzymes or ubiquitin itself could be modified as a free species before catalytic involvement. Post-translational modification prior to Ub engagement could lead to a variety of consequences, including altered Ub activation or conformational changes to enzyme structure, for example. Secondly, post-translational modifications could target a transient Ub-containing complex, namely the thioester intermediates formed between E1~Ub, E2~Ub, or E3~Ub. Here, the PTM might occur on either the enzyme or the Ub molecule and could be expected to have different downstream effects. Modification at this step in the cascade would likely influence complex stability or conformation. Finally, the proteins involved in ubiquitination could undergo modification after Ub has been cycled through the cascade. PTMs at this point in the cycle might inhibit recurring catalysis, limiting the function of the E1, E2, or E3 enzymes. Modification of Ub, once contained in a polyUb chain (either free or substrate-bound), could switch chain topology, limit processing by other proteins (such as the proteasome) or serve as a distinct signal to alter protein interactions.

In this review, we have highlighted how acetylation, phosphorylation, and ubiquitination PTMs might act in activating and inhibiting manners, and how existing structures of the enzymes might be used to rationalize the outcomes of PTMs during ubiquitination. In addition to the common PTMs described here, several ubiquitination proteins undergo less common PTMs such as deamidation [192,193], or phosphoribosylation [194], which also have effects on their activity. The complexity of PTM crosstalk in disease is ever expanding, as we identify new disease-causing organisms, new proteins with the capability of modifying others, new target sites on pre-existing substrates, and potential new post-translational modifications that regulate protein function.

## Figures and Tables

**Figure 1 biomolecules-12-00467-f001:**
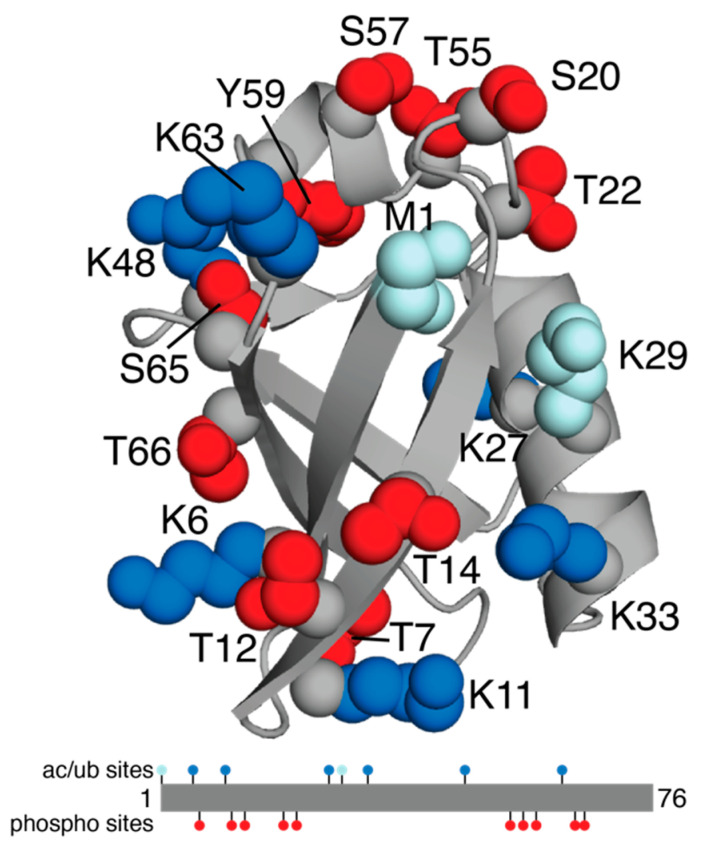
Ubiquitin post-translational modifications. The cartoon structure of Ub (PDB 1UBQ) [91] is shown in grey with PTM sites indicated: ubiquitination only (pale blue), acetylation or ubiquitination (blue), and phosphorylation (red). Also shown is the linear structure of Ub with the PTMs.

**Figure 2 biomolecules-12-00467-f002:**
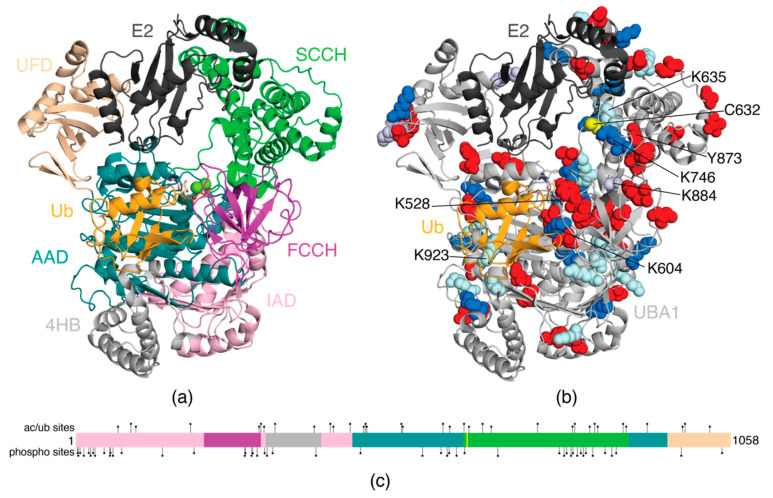
Cartoon model of the human E1 enzyme UBA1 in complex with the E2 conjugating enzyme, Ubc4 and ubiquitin. (**a**) Domains in UBA1 (PDB 6DC6) are indicated as 4HB (light grey), IAD (light pink), AAD (teal), FCCH (magenta), SCCH (green), UFD (wheat). The E2 enzyme (dark grey) and bound ubiquitin (orange) at the adenylation site are also shown. The E2 enzyme was modeled based on PDB coordinates 4II2 [98] and occupies a position consistent with other E1:E2 structures [99,100,101]. (**b**) The cartoon structure of UBA1 is shown in light grey, and the locations of acetylation or ubiquitination (blue), acetylation only (light purple), ubiquitination only (pale blue), and phosphorylation (red) sites in UBA1 are indicated. The catalytic cysteine (C632) is shown in yellow, and the E2 and Ub-adenylate are coloured as in (**a**). Only those PTM sites discussed in detail in the text are labelled here. (**c**) Linear domain organization of human UBA1 with colours matched to (**a**). The catalytic cysteine is indicated by a yellow bar, and all observed PTM sites in human UBA1 are indicated to the top (acetylation or ubiquitination) and bottom (phosphorylation) of the schematic.

**Figure 3 biomolecules-12-00467-f003:**
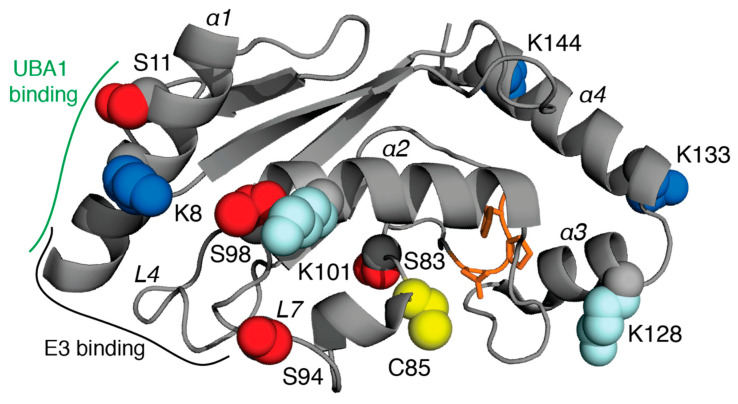
Acetylation, phosphorylation, and ubiquitination sites in E2 conjugating enzymes. Residues observed to be acetylated or ubiquitinated (blue), phosphorylated (red), or only ubiquitinated (pale blue) in the UBE2D family (UBE2D1-UBE2D4) are indicated using the structure of UBE2D3 (PDB 5EGG) as a template [118]. Not all sites are found in all family members. The catalytic cysteine (C85) is shown in yellow and the HPN motif is shown in orange sticks. Helices and discussed loops are labelled in italics. Regions along helix α1 and the L4, L7 loops where UBA1 and E3 enzymes interact are indicated.

**Figure 4 biomolecules-12-00467-f004:**
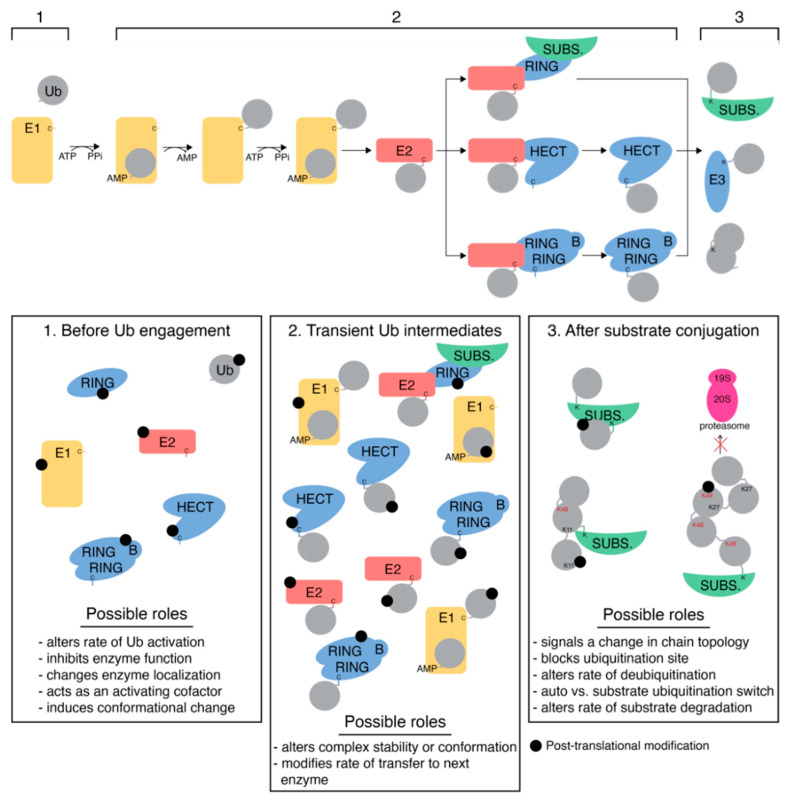
Implications of post-translational modification of ubiquitination proteins. The ubiquitination pathway for RING, HECT, and RBR E3 ligases is shown at the top. Post-translational modification of these proteins can occur at three points in the ubiquitination cycle: (1) before Ub engagement, (2) contained in a transient Ub complex, or (3) after substrate conjugation. Thioester intermediates are indicated by ‘C’ (cysteine), while stable isopeptide linkages are denoted by ‘K’ (lysine). Potential roles of PTMs at each point in the cascade are listed.

**Table 1 biomolecules-12-00467-t001:** Post-translational modification sites in human ubiquitination proteins ^#^.

Gene	Protein	UniProt	Acetylation ^+^	Phosphorylation ^+^	Ubiquitination ^+^	Disease
Ubiquitin						
UBC	Ubiquitin	P0CG48	K6, K11, K27, K33, K48, K63	T7, T12, T14, S20, T22, T55, S57, Y59, S65, T66	M1, K6, K11, K27, K29, K33, K48, K63	Multiple myeloma, leukemia, colorectal cancer
E1 Activating Enzyme						
UBA1	UBE1	P22314	K68, K89, K185, K385, K465, K470, K526, K528, K593, K604, K627, K657, K671, K746, K838, K843, K884, K980, K984, K1024	S3, S4, S7, S13, S21, S24, S28, S31, S46, Y55, S56, Y60, S74, S140, T191, Y273, T274, S284, Y286, S293, S305, S309, T318, Y388, S460, Y560, Y590, T600, T603, T615, S628, T682, S781, T789, S793, T800, S803, S810, S816, S820, S824, S835, S855, S866, Y873, Y978, S1044	K68, K89, K97, K185, K296, K299, K304, K322, K385, K411, K416, K443, K465, K468, K470, K526, K528, K593, K604, K627, K635, K657, K671, K746, K802, K806, K830, K838, K843, K889, K923, K980	Melanoma, lymphoma, leukemia, breast, colorectal, and gastric cancers
E2 Conjugating Enzymes						
UBE2A	UBE2A, RAD6A	P49459	K66	S120, S142, S148	K66, K75	Breast, lung cancer

UBE2B	UBE2B, RAD6B	P63146	K66	T69, S120	K66	Leukemia

UBE2C	UBE2C, UbcH10	O00762	K18, K119, K121, K164	S3, T11, S51, S87, T160	K80, K119, K121, K164, K172	Leukemia

UBE2D1	UbcH5a	P51668	K144	S83, Y145	K144	Lung cancer

UBE2D2	UbcH5b	P62837	K8, K144	S83, S94, T98	K8, K101, K128, K144	Lung cancer

UBE2D3	UbcH5c	P61077	K8, K133, K144	S11, S83, S94, T98	K8, K101, K128, K133, K144	Breast, lung cancers, leukemia

UBE2D4	UbcH5d	Q9Y2X8	K8, K144		K144	Leukemia, multiple myeloma

UBE2E1	UbcH6	P51965	K43, K50, K136	S2, S6, S9, T28, S46, Y77	K24, K43, K50, K54, K72, K136	Breast, colorectal cancers, leukemia

UBE2E2	UbcH8 (ISG15)	Q96LR5	K48, K52	T3, S11, S13, T14, S15, S18, S19, T49, S54, Y85	K62, K144	Breast, colorectal, lung, gastric cancers, leukemia

UBE2E3	UbcH9	Q969T4	K50, K58	S3, S8, S12, S19, Y91	K39, K58, K68, K150	Breast, lung, gastric cancers, leukemia

UBE2F	UBE2F, NCE2 (NEDD8)	Q969M7	K7, K9	S31, T85, S124, Y179	K7	Breast, brain, lung, gastric cancers

UBE2G1	Ubc7	P62253		T2, S6, Y65, T76, Y102, Y104	K19, K63, K73, K101, K106, K163	Multiple myeloma

UBE2G2	Ubc7	P60604			K7, K142, K153, K156, K161	Leukemia

UBE2H	UbcH2, E2-20K	P62256	K8, K60, K64, K147	S2, S3, S5, T13, S65, S166, S169	K17, K60, K64, K147	Breast cancer

UBE2I	Ubc9 (SUMO)	P63279	K30, K48, K59, K65, K74, K146	T35, S70, S71	K18, K49, K59, K65, K74	Bone cancer, leukemia

UBE2J1	NCUBE1	Q9Y385	K8, K17	Y5, S9, S51, S184, S251, S266, T267, S268, T282, T295, Y307, Y312	K8, K143, K164, K177, K186, K194	Breast, lung cancers, leukemia, multiple myeloma


UBE2J2	NCUBE2	Q8N2K1	K18	Y31, Y46	K18, K64, K139, K152, K154, K168	Esophageal cancer

UBE2K	E2-25K, HIP-2	P61086	K14, K18, K72, K142, K164, K165	T49, S158 *, S159, Y162 *, T163 *, S185	K14, K18, K24, K28, K61, K72, K97, K142, K164, K165	Leukemia, lung cancer

UBE2L3	UbcH7	P68036	K9, K20, K64, K73, K82, K96, K131, K138, K145	Y75, Y129, Y147	K9, K20, K48, K64, K67, K71, K73, K82, K96, K100, K131, K138, K145, K150	Lung, colorectal cancers, lymphoma, leukemia

UBE2L6	UbcH8 (ISG15)	O14933	K138	S26, S153	K9, K16, K17, K96, K138	Breast cancer, leukemia, neuroblastoma

UBE2M	Ubc12 (NEDD8)	P61081	K3, K8, K36, K45, K72	S6, T20, S23, S28, T46, S50, S52 *, Y86, Y172, Y177	K3, K8, K26, K36, K45, K61, K72, K75, K81, K92, K94	Breast, lung cancers, leukemia, lymphoma

UBE2N	Ubc13	P61088	K10, K24, K53, K82, K92, K94	Y34, S45, T139, T144, Y147	K10, K24, K68, K74, K82, K92, K94	Breast, lung, gastric, colorectal cancers, leukemia, lymphoma

UBE2O	E2-230K, KIAA1734	Q9C0C9	None in UBC	None in UBC	K953, K990, K1038	Leukemia, lymphoma

UBE2Q1	UBE2Q, NICE5	Q7Z7E8	K403	Y264, S391, Y393, S394, S401, S404, Y415	K307, K390, K403	Breast cancer

UBE2Q2	UBE2Q2	Q8WVN8		S357, Y368, T369		Breast, gastric cancers

UBE2R1	Cdc34, UBCH3B	P49427	K167, K173	Y68, S71, T89, T162, Y190, S203, S222, S231, T233, S236	K11, K18, K63, K157, K167, K182, K193	Bone, colorectal, cervical cancers

UBE2R2	Cdc34B, UBC3B	Q712K3		Y190, S202, Y207, Y228, S233, S238	K11, K18, K63, K157, K159, K167, K173, K182, K193, K195	Leukemia

UBE2S	UBE2S, E2-24K	Q16763	K68, K82	S73, Y78, T81, T152, S173, S175, T180	K18, K32, K63, K68, K76, K100, K117, K197, K198	Prostate, gastric cancers, leukemia

UBE2T	UBE2T	Q9NPD8	K28, K91, K191	T72, S172, S177, T178, S184	K28, K48, K91, K136, K141, K156, K182, K191, K192	Breast, lung cancers, leukemia, multiple myeloma

UBE2U	UBE2U	Q5VVX9		None in UBC		

UBE2V1	UEV1	Q13404	K10, K24, K30, K74	S7, T86, S106, Y145, S146	K10, K68, K74, K131	Breast, lung cancers, leukemia

UBE2V2	UEV2, MMS2	Q15819	K8, K66, K72, K108	S4, T5, S79, S102	K8, K66, K72, K108, K129, K133	Breast, colorectal cancers, leukemia

UBE2W	UBC16	Q96B02		S29, S33	K10	

UBE2Z	UBE2Z, USE1(FAT10)	Q9H832	K166, K238	None in UBC	K113, K166, K238	Leukemia

^#^ Post-translational modification sites have been curated from references [8,9,10,52,53,54,55,56,57,58,59,60,61,62,63,64,65,66,67,68,69,70,71,72,73,74,75,76,77,78,79,80,81,82,83,84,85,86,87,88,89,90]. For specific sites readers are encouraged to review data in the original references. ^+^ Only post-translational modifications in the UBC fold region are listed. In many cases there are extensive modifications in accessory regions/domains. * Some sites have low probabilities.

## Data Availability

Most of the data presented in this study are openly available in PhosphoSitePlus^®^ at doi: 10.1093/nar/gku1267, reference number [107] and in UniProt at doi: 10.1093/nar/gkaa1100, reference number [195]. Data not contained in PhosphoSitePlus^®^ or UniProt are available in the supplemental materials for references [8,9,10,52,53,54,55,56,57,58,59,60,61,62,63,64,65,66,67,68,69,70,71,72,73,74,75,76,77,78,79,80,81,82,83,84,85,86,87,88,89,90].

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
