# Peer review of "Acetylation, Phosphorylation, Ubiquitination (Oh My!): Following Post-Translational Modifications on the Ubiquitin Road"

_biomolecules, 2022, doi:10.3390/biom12030467_

Round 1
Reviewer 1 Report
I’d be happy to provide a review of the manuscript entitled “Acetylation, phosphorylation, ubiquitination (oh my!): Following post-translational modifications on the ubiquitin road” by Lacoursiere et al submitted to Biomolecules. This review described in great detail the importance of some post-translational modifications including acetylation, phosphorylation and ubiquitination on ubiquitin and other enzymes of the ubiquitination pathway. The authors initially described the various modifications and their functional significance to ubiquitin, the master tag of the ubiquitination pathway. The fact that ubiquitin is modified by post-translationally was “recently” discovered. The best studied modification of ubiquitin is Ser65 and its critical role in the activation of Parkin was described. Additional roles for acetylation of some Lys residues on ubiquitin were also discussed. The description of the post-translational modifications continued with the three enzyme families that are responsible for attaching ubiquitin to substrate proteins. A significant analysis of E1 and the functional/ catalytic effects of PTMs in its different domains resulting in the formation of E1-Ub was described. The concluding paragraph to this section on E1 was very informative and made the reviewer aware of the large number of possible sites of regulation of enzymatic activity.
With almost 40 E2 enzymes, with distinct sequences in their UBC cores, it’s not surprising to see that there is a concentration of PTMs occurring within close proximity to the sites of interaction with E1s or E3s which would represent two areas that should be highly regulated. It is also intriguing that there seems to be a single hotspot residue that regulates activity in at least some but not all E2s.
With over 1000 distinct E3 ligases, it is not surprising to see that PTMs have such a significant impact on regulation of ubiquitination using a variety of different mechanisms. Although the authors chose to describe only a few E3 ligases, I was disappointed to only see phosphorylation. It is well known that MDM2 and UHRF1 activities are also regulated by acetylation and I would like to see this added to the manuscript.
Specific points.
Figure 0: Can the authors add the linear and 3D structure of Ub along with the PTMs especially Ser65.
Figure 1: Can a schematic diagram of the linear E1 protein be included to permit the readers to better visualize the corresponding domains. The E2 and Ub molecules are not easy to distinguish.
Can the authors explain or elaborate why E2 PTMs are more prevalent in diseases?
Figure X: Is there any way to include a figure for E3 ligases?
Figure Y: Can there be a concluding model to aid in the visualization of the scope of the manuscript.
Author Response
Thank you for the excellent comments. We have addressed all of these as follows.
Figure 0: Can the authors add the linear and 3D structure of Ub along with the PTMs especially Ser65.
We have added this as new Figure 1
Figure 1: Can a schematic diagram of the linear E1 protein be included to permit the readers to better visualize the corresponding domains. The E2 and Ub molecules are not easy to distinguish.
We have added a linear E1 protein that schematically shows all the PTMs, now Figure 2. This has been coloured the same as the structure in Figure 2a
Can the authors explain or elaborate why E2 PTMs are more prevalent in diseases?
At present this is difficult due to lack of data. We have tried to correlate with structures as best we could.
Figure X: Is there any way to include a figure for E3 ligases?
Ideally this should be included. However since most PTMs lie outside the RING/HECT regions, and there is a great deal of variation in the sequences/domains of E3 ligases we thought it is best to leave this out.
Figure Y: Can there be a concluding model to aid in the visualization of the scope of the manuscript.
We have added a new concluding model, Figure 4, and a new paragraph in the Conclusions. This was an excellent idea.
Reviewer 2 Report
The review by Lacoursiere et al. aims to provide an exhaustive overview of post-translational modifications of ubiquitin and ubiquitin pathway enzymes, including E1, E2, and E3 ligases. The authors review the results of high-throughput and more targeted investigations and put them into context with medical outcomes. An exhaustive table of PTMs and the associated diseases curated from the literature is provided which provides a highly useful resource to the field (and could be even more useful if the sources were cited not summarily at the bottom, but with each individual row of data). PTMs in E1 and E2 enzymes are presented mapped to atomic structures to visualize their impact, but this level of detail is not provided for PTMs on E3 enzymes. The review is very well written and presents as is a wide and complicated field in a concise and structured manner.
I strongly recommend publication, with some very minor edits:
78-79: this sentence reads a bit awkward, possibly rephrase it.
144: layout
459: missing comma?
Figure 2: The UBE2D family consists of four near-identical enzymes that differ only in few residues that are replaced in a very conserved fashion. I fully agree that the structures are expected to be near-identical, furthermore the sequences align without gaps, therefore it is correct to show a single structure in the figure representing UBE2D1-4. I would nevertheless recommend to mention which structure (PDB code) is presented here.
Author Response
Thank you for the very supportive comments. We have addressed the concerns described below.
78-79: this sentence reads a bit awkward, possibly rephrase it.
This has been modified
144: layout
459: missing comma?
We could not find these issues
Figure 2: The UBE2D family consists of four near-identical enzymes that differ only in few residues that are replaced in a very conserved fashion. I fully agree that the structures are expected to be near-identical, furthermore the sequences align without gaps, therefore it is correct to show a single structure in the figure representing UBE2D1-4. I would nevertheless recommend to mention which structure (PDB code) is presented here.
The PDB code and reference have been added.